# Electrocorticographic dissociation of alpha and beta rhythmic activity in the human sensorimotor system

Arjen Stolk[1,2]*, Loek Brinkman[3], Mariska J Vansteensel[3], Erik Aarnoutse[3], Frans SS Leijten[3], Chris H Dijkerman[4], Robert T Knight[1], Floris P de Lange[2], Ivan Toni[2]

[1]Helen Wills Neuroscience Institute, University of California, Berkeley, Berkeley, United States; [2]Donders Institute for Brain, Cognition, and Behaviour, Radboud University, Nijmegen, Netherlands; [3]Department of Neurology and Neurosurgery, UMC Utrecht Brain Center, UMC Utrecht, Utrecht, Netherlands; [4]Helmholtz Institute, Experimental Psychology, Utrecht University, Utrecht, Netherlands

**Abstract** This study uses electrocorticography in humans to assess how alpha- and beta-band rhythms modulate excitability of the sensorimotor cortex during psychophysically-controlled movement imagery. Both rhythms displayed effector-specific modulations, tracked spectral markers of action potentials in the local neuronal population, and showed spatially systematic phase relationships (traveling waves). Yet, alpha- and beta-band rhythms differed in their anatomical and functional properties, were weakly correlated, and traveled along opposite directions across the sensorimotor cortex. Increased alpha-band power in the somatosensory cortex ipsilateral to the selected arm was associated with spatially-unspecific inhibition. Decreased beta-band power over contralateral motor cortex was associated with a focal shift from relative inhibition to excitation. These observations indicate the relevance of both inhibition and disinhibition mechanisms for precise spatiotemporal coordination of movement-related neuronal populations, and illustrate how those mechanisms are implemented through the substantially different neurophysiological properties of sensorimotor alpha- and beta-band rhythms.
DOI: https://doi.org/10.7554/eLife.48065.001

*For correspondence:
astolk@berkeley.edu

## Introduction

To control a movement, specific neuronal populations supporting particular features of that movement need to be facilitated while other populations need to be suppressed (*Ebbesen and Brecht, 2017*; *Greenhouse et al., 2015*; *Mink, 1996*). Both operations need to be organized in a precise spatiotemporal pattern, such that the demands of coordinating body segments for movement are dynamically solved through the selective excitation and inhibition of relevant and irrelevant sensorimotor neuronal populations, respectively (*Bruno et al., 2015*; *Dombeck et al., 2009*; *Graziano, 2016*; *Shenoy et al., 2013*). One putative mechanism through which this sensorimotor coordination is implemented is the rhythmic modulation of neuronal local field potentials in the alpha (8–12 Hz) and beta (15–25 Hz) frequency range (*Brovelli et al., 2004*; *Pfurtscheller and Berghold, 1989*; *Picazio et al., 2014*; *van Wijk et al., 2012*).

Neuronal local field potentials in the sensorimotor cortex are organized in two prominent spectral clusters, alpha- and beta-band rhythms, known to be relevant for movement selection and to differ across several features. For instance, there are differences in the cortico-subcortical loops supporting alpha- and beta-band rhythms (*Bastos et al., 2014*; *Leventhal et al., 2012*; *Schreckenberger et al., 2004*; *West et al., 2018*), and only the latter rhythm has clear modulatory effects on corticospinal

neurons (*Baker et al., 1997*; *Madsen et al., 2019*; *Mima and Hallett, 1999*; *van Elswijk et al., 2010*). Yet, the neurophysiological characteristics of alpha- and beta-band rhythms have often been studied by aggregating these two rhythms into the same (mu-) rhythm category (*Cuevas et al., 2014*; *Hari, 2006*; *Miller et al., 2010*), an approach often justified by the partial overlap in their spatial and spectral distributions (*Bressler and Richter, 2015*; *Haegens et al., 2014*; *Salmelin and Hari, 1994*; *Szurhaj et al., 2003*) and by the temporal correlation of their power envelopes (*Carlqvist et al., 2005*; *de Lange et al., 2008*; *Tiihonen et al., 1989*). By aggregating those rhythms, it has been recently shown that 4–22 Hz activity modulates high-frequency broadband power in primates' frontal cortex (*Bastos et al., 2018*; *Johnston et al., 2019*), and that 10–45 Hz activity is spatially organized in traveling waves (*Rubino et al., 2006*; *Takahashi et al., 2015*). It remains unclear, however, whether that aggregation could obscure differential contributions of those rhythms to movement selection. For instance, it is an open question whether alpha- and beta-band rhythms modulate the excitability of the same neuronal ensembles in the same direction when a movement is selected across the sensorimotor cortex (*Brinkman et al., 2016*; *Brinkman et al., 2014*).

Here we used direct recordings from the human cortical surface (electrocorticography, ECoG; *Figure 1A*) to assess the anatomical and functional specificity of alpha- and beta-band rhythms and their effects on the local excitability of sensorimotor neuronal ensembles during performance of a motor imagery task that offers a window into movement selection. Local cortical effects were quantified through two complementary power-spectral metrics of excitability. First, we considered high-frequency (60–120 Hz) content in the ECoG signal, a mesoscale correlate of action potentials and dendritic currents in the local neural population (*Leszczynski et al., 2019*; *Manning et al., 2009*; *Miller et al., 2009*; *Ray and Maunsell, 2011*; *Rich and Wallis, 2017*). Second, we considered the slope of the power-spectral density function ($1/f$ slope), a putative summary index of synaptic excitation/inhibition balance (*Gao et al., 2017*). Furthermore, rather than assuming that alpha- and beta-band rhythms are spatially stationary across the sensorimotor cortex (*Brinkman et al., 2016*; *Brinkman et al., 2014*), we examined the spatio-temporal distribution of the two sensorimotor rhythms and their cortical effects through two complementary analyses. First, we considered the organization of spatially systematic phase relationships among rhythmic signals (traveling waves) across the sensorimotor cortex (*Ermentrout and Kleinfeld, 2001*; *Muller et al., 2018*). Second, we explored the spatiotemporal relation between rhythm strength and local cortical excitability through analysis of representational similarity between those spectral markers (*Kriegeskorte et al., 2006*).

This neurophysiological characterization of alpha- and beta-band rhythms is based on a

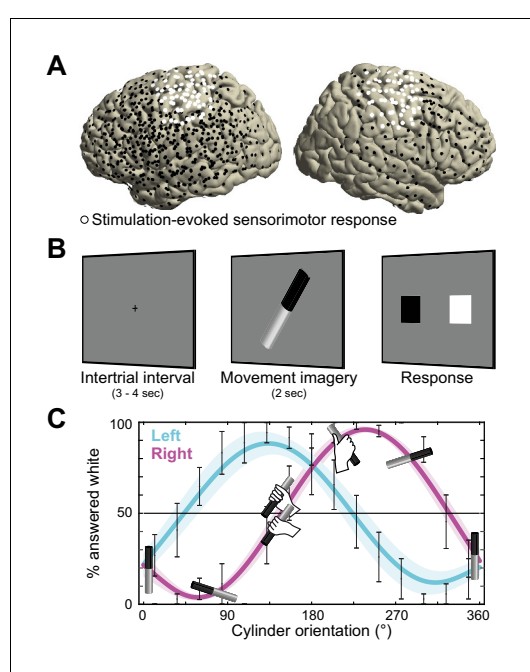

**Figure 1.** Recording electrode locations and movement imagery task. (**A**) Neural signals were recorded from the cortical surface of eleven epilepsy patients that were implanted with subdural electrode grids and strips. The electrode locations of all participants are overlaid on a template brain (black markers). Electrodes resulting in either a somatomotor or somatosensory response in the upper limb upon electrical stimulation are highlighted in white. (**B**) Participants imagined grasping the middle-third of a black-white cylinder with either their left or right hand. At the response screen, they indicated whether their thumb was on the black or the white part of the cylinder at the end of the imagined movement. (**C**) The preferred manner in which the cylinder was grasped (thumb on black or white part, related to overhand vs. underhand grasping) was modulated as a function of the cylinder's orientation and differed for the left and right hand. Error bars indicate M ± SEM over nine participants. Lines and shaded areas indicate M ± SEM of sine-wave fits to individual over-/underhand data points.

DOI: https://doi.org/10.7554/eLife.48065.002

principled differentiation of the two sensorimotor rhythms along spectral, anatomical, and movement-related dimensions. Spectrally, alpha- and beta-band signals were disambiguated from arrhythmic spectral components in each individual participant (*Wen and Liu, 2016*). This procedure increases spectral precision and physiological interpretability by controlling for the effects of task-related power-spectral 1/*f* modulations over those rhythms (*He, 2014*). Anatomically, the ECoG recordings were precisely registered to the cortical anatomy of each patient (*Stolk et al., 2018*), and sorted according to the sensorimotor responses evoked by electrical stimulation of the electrodes. Functionally, the movement-related specificity of alpha- and beta-band signals was experimentally controlled by using imagined movements psychophysically-matched to actual movements (*Figure 1B*; *Brinkman et al., 2014*; *Rosenbaum et al., 1995*). This procedure is grounded on shared processes between movement selection and motor imagery. Besides sharing motor control variables and sensitivity to biomechanical constraints (*de Lange et al., 2006*; *Gentili et al., 2004*; *Vargas et al., 2004*), movement selection and motor imagery evoke the same activity patterns in dorsal premotor cortex and in the subthalamic nucleus (*Cisek and Kalaska, 2004*; *Kühn et al., 2006*), leading to similar consequences on the excitability of the corticospinal system (*Lebon et al., 2019*). Moreover, using motor imagery increases functional interpretability by avoiding confounding execution-related somatosensory reafference known to differentially affect post-movement power dynamics in the alpha- and beta-bands (*Alayrangues et al., 2019*; *Jurkiewicz et al., 2006*; *Tan et al., 2016*).

## Results

### Direct cortical recording during psychophysically-controlled movement imagery

Neurosurgical epilepsy patients implanted with subdural grid and strip electrode arrays for clinical diagnostic purposes performed up to three sessions of a movement imagery task where they imagined how to grasp an object with either their left or right hand. Eleven patients participated, eight with left hemisphere arrays, and three with arrays on the right (see overlay on a template brain in *Figure 1A*). Two participants experienced difficulties adhering to the task instructions and were excluded from further analysis.

The motor imagery task involved 60 trials per session. Each trial started with the presentation of a black-white cylinder on a computer screen. Participants imagined how to grasp the middle-third of that cylinder with either their left or right hand, in alternating blocks of 10 trials (*Figure 1B*). After a fixed amount of time, a response screen appeared where the participants indicated whether their thumb was on the black or the white part of the cylinder at the end of the imagined movement. The response screen consisted of two squares on the horizontal plane (one black and one white), where participants indicated 'black' or 'white' by pressing the corresponding button using their left or right thumb on a button box that they held with both hands. The relative location of the black and white squares on the screen was pseudo-randomized across trials to prevent the preparation of the thumb response during the simulation of the grasping movements.

The task was designed to assess whether participants produced imaginary movements conforming to the biomechanical constraints of the corresponding real movements. On each trial, the cylinder was pseudo-randomly tilted according to 1 of 15 possible orientations, spanning 0 to 360°. This task manipulation resulted in trials affording both overhand and underhand grasping, and trials that afforded grasping in a single manner only due to biomechanical constraints of the hand. As seen in *Figure 1C*, the preferred manner in which participants imagined grasping the cylinder (thumb on black or white part) depended on the orientation of the cylinder and followed the biomechanical constraints of the body. This is supported by a psychophysical analysis showing that a sine-wave fit to the over-/underhand data points explained $81 \pm 4\%$ of the variance in the left-hand condition (M ± SEM; $t(8) = 18.4$, $p<0.001$) and $76 \pm 4\%$ in the right-hand condition ($t(8) = 21.6$, $p<0.001$), consistent with the prediction of two orientation-dependent switch points in each hand's response curve, that is the 50% crossings in *Figure 1C* (*Brinkman et al., 2014*).

Eight out of nine participants additionally completed a control task that used the same visual input and response contingencies as the motor imagery task, but where no imagery was required. In the control task, the surface areas of the cylinder differed slightly across trials, for example 54%

black and 46% white, and participants reported which side of the black-white cylinder was larger. This allowed correcting for neural changes unrelated to the movement imagery process, such as those evoked by the visual input. Participants performed the control task with high accuracy (99.4 ± 0.3% correct, M ± SEM).

In the following sections, we first characterize the anatomical distribution and task-related temporal profile of neuronal ensembles supporting alpha- and beta-band rhythms across the sensorimotor cortex, as well as the functional consequences of electrical stimulation of those ensembles. Afterward, we assess the influence of those rhythms on the spatiotemporal pattern of sensorimotor excitability during imagined movement and the spatiotemporal organization of those rhythms across the sensorimotor cortex.

## Alpha- and beta-band rhythms build on anatomically distinct neuronal ensembles

Neuronal ensembles producing sensorimotor alpha- and beta-band rhythms across the human sensorimotor cortex were isolated with a four-step procedure. The goal of the procedure is to characterize the spatial distribution of rhythmic and spectrally homogeneous neural activity in sensorimotor areas in each participant's subdural grid and strip electrode arrays.

First, for each participant, we selected electrodes that upon electrical stimulation yielded somatomotor or somatosensory responses of the upper limb contralateral to the cortical grid (i.e., twitches, movements, tingling of fingers, hand, wrist, arm, or shoulder). This procedure identified cortical regions supporting sensorimotor components of movement (white electrodes in *Figures 1A* and *2A*). Seven out of nine participants showed such responses, indicating sensorimotor coverage in these participants. Second, we used irregular-resampling auto-spectral analysis (IRASA, *Wen and Liu, 2016*) of the neural signal recorded at the stimulation-positive electrodes. This procedure isolated specific rhythmic activity embedded in the concurrent broadband 1/$f$ modulations. Third, mean and full-width at half-maximum of alpha and beta spectral distributions were defined for each participant using a Gaussian model (red and blue areas of the power-spectra in *Figure 2A*). This adaptive approach (*Source code 1*) avoids having to rely on canonical frequency bands that may not accurately capture the neural phenomena of interest in each individual (*Haegens et al., 2014*; *Szurhaj et al., 2003*). Five out of seven participants had a rhythmic power-spectral component that overlapped with the 8–12 Hz alpha frequency range, one had a rhythmic component below that range, and all seven had a rhythmic component that overlapped with the 15–25 Hz beta range (*Figure 2—figure supplement 1*). Participant S7 exhibited only a single rhythmic component (in the beta frequency range) and was excluded from further analysis. On average, the remaining six participants' alpha and beta frequency bands were centered on 7.4 ± 0.7 and 16.9 ± 1.1 Hz (M ± SEM), respectively. Fourth, we localized cortical sites showing relative maxima in alpha and beta power. We selected electrodes that exceeded the upper limit of the 99% confidence interval for absolute spectral power in the respective frequency band across all stimulation-positive electrodes defined by the first step. This analysis yielded 4.0 ± 1.2 alpha and 3.4 ± 0.8 beta peak activity electrodes for participants S1 - S5 (M ± SEM, red and blue electrodes in *Figures 2A* and *1*). Due to limited sensorimotor coverage, the number of electrodes could not be narrowed down for participant S6, and the four stimulation-positive electrodes in this participant were used for the analysis of temporal dynamics only.

The cortical sites isolated through this principled four-step procedure had systematically different functional and anatomical properties. All 20 electrodes with alpha-band local maxima were located posterior to the central sulcus, $\chi^2(19)=40$, p<0.001 (pre vs. postcentral sulcus), see the red electrodes in *Figure 2D*. As seen in the same figure, the 17 blue electrodes with beta-band local maxima were localized to both sides of the central sulcus, $\chi^2(16)=1.1$, p=0.3 (7 pre- and 10 postcentral). Furthermore, only 7 out of 30 combined unique electrodes were local maxima for both sensorimotor rhythms, suggesting that alpha- and beta-band rhythms involve largely different neuronal ensembles, $\chi^2(29)=17$, p<0.001. On average, alpha- and beta-band local maxima were separated by 11.8 ± 2.2 mm (M ± SEM).

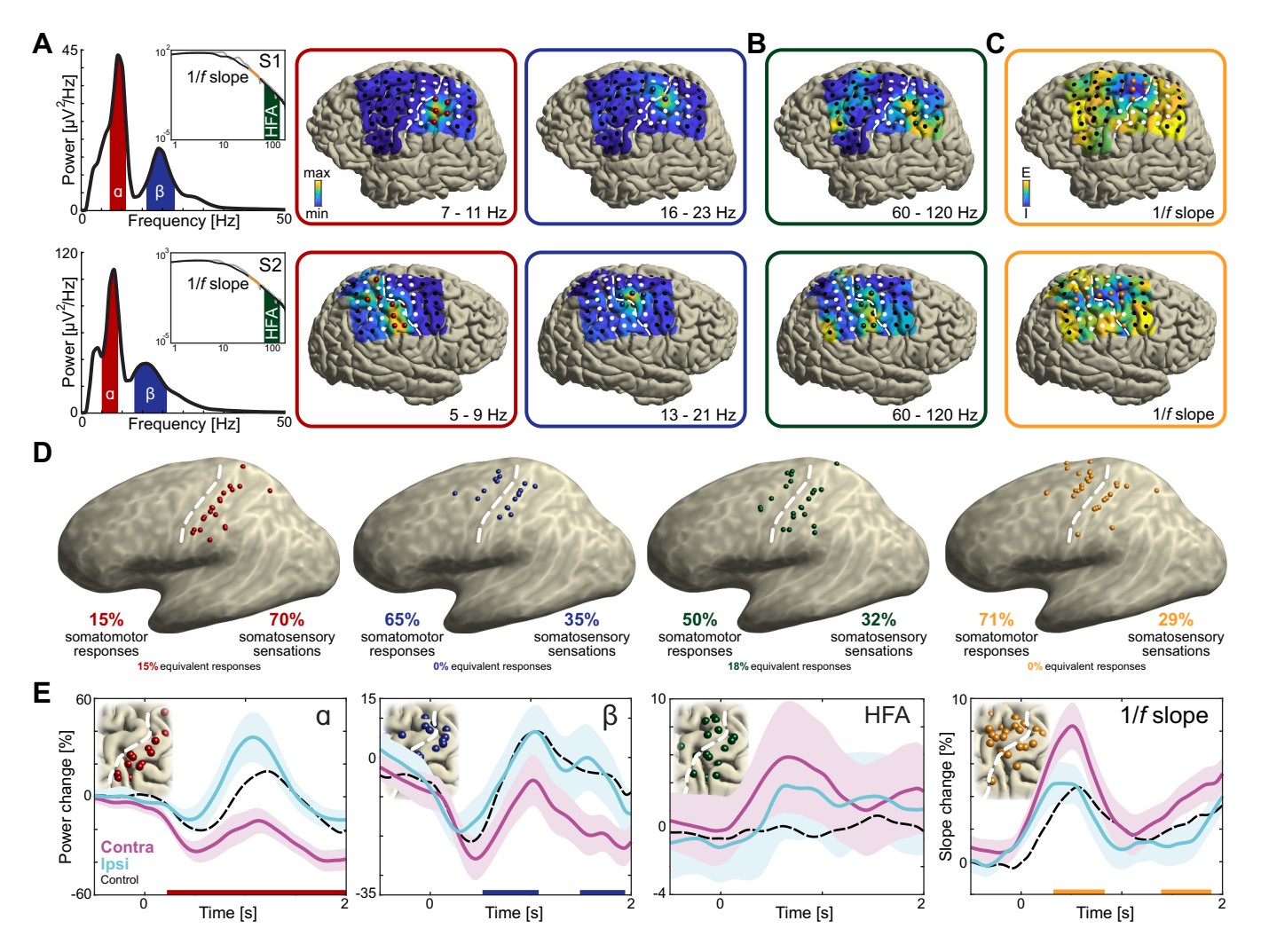

**Figure 2.** Anatomical and functional dissociation of sensorimotor alpha and beta. (**A**) Spectral and spatial distributions of alpha and beta rhythmic activity during imagined movement in two representative individuals. The insets show in log-log space the original power-spectra (in gray) and extracted arrhythmic $1/f$ content (black) that gave rise to the participant-specific rhythmic content shown in the main graph on the left. The color axes of the cortical maps run from minimum in blue to maximum absolute spectral power in yellow. White electrodes yielded somatomotor or somatosensory responses of the upper limb following electrical stimulation. Red and blue electrodes represent alpha- and beta-band local maxima across the sensorimotor cortex, respectively. (**B**) As the cortical maps in A, but for 60 to 120 Hz high-frequency arrhythmic content (HFA) of the ECoG signal. Green electrodes represent high-frequency-band local maxima across the sensorimotor cortex. (**C**) Ditto, but for the $1/f$ slope between 30 and 50 Hz, indicated by the orange graph sections in the insets of A. The $1/f$ slope is a putative power-spectral index of synaptic excitation/inhibition balance. Orange electrodes represent sensorimotor sites with relatively the strongest inhibition, that is the steepest slope. (**D**) Template brains showing the local maxima from five individuals visualized on the left hemisphere. Alpha is maximal at electrodes on the postcentral gyrus that yielded somatosensory sensations of the upper limb following electrical stimulation (red electrodes). In contrast, beta is strongest at electrodes placed over the central sulcus, with electrical stimulation yielding both movements and somatosensory sensations (blue electrodes). White dashed lines indicate central sulci. (**E**) Temporal dynamics of power changes aggregated across the relevant local maxima during imagined movement of the contralateral or ipsilateral arm. Both neuronal ensembles producing alpha and beta rhythms showed effector-specific modulation during motor imagery, from 0 to 2 s. Shaded areas indicate ±1 SEM. Colored bars along the x-axes indicate time intervals of statistically significant lateralization effects. Dashed black lines represent mean activity in the control task, for reference.

DOI: https://doi.org/10.7554/eLife.48065.003

The following figure supplements are available for figure 2:

**Figure supplement 1.** As in *Figure 2A–C*, for seven individuals with sensorimotor coverage.

DOI: https://doi.org/10.7554/eLife.48065.004

*Figure 2 continued on next page*

*Figure 2 continued*

**Figure supplement 2.** Single-trial broadband/unfiltered cortical signals from alpha- and beta-band local maxima (highlighted red and blue electrodes, respectively) in two representative individuals.

DOI: https://doi.org/10.7554/eLife.48065.005

**Figure supplement 3.** As in *Figure 2E*, but with mean temporal dynamics of high demand trials (solid lines, cylinder orientations that afforded both overhand and underhand grasping) and low demand trials (dashed lines, cylinder orientations that afforded grasping in a single manner only).

DOI: https://doi.org/10.7554/eLife.48065.006

## Alpha- and beta-band rhythms build on neuronal ensembles with different sensorimotor properties: effects of electrical stimulation

To test whether the neuronal ensembles generating alpha and beta rhythms had different functional properties, we probed the somatosensory and motor responses evoked by electrical stimulation of those ensembles. As indicated in *Figure 2D*, alpha electrodes yielded predominantly (14 out of 20 electrodes, 70%) somatosensory sensations of the contralateral upper limb following electrical stimulation, $\chi^2(19)=12.4$, p<0.001. Additionally, a subset of electrodes (3 out of 20, 15%) were part of equally many stimulation electrode pairs yielding both somatomotor and somatosensory responses. These observations suggest that alpha activity predominantly supports somatosensory components of a movement, in line with its anatomical distribution along the postcentral gyrus. By contrast, beta electrodes were marginally more likely (11 out of 17, 65%) to elicit a somatomotor than a somatosensory response of the upper limb following electrical stimulation, $\chi^2(16)=2.9$, p=0.086.

## Alpha- and beta-band rhythms contribute to movement imagery with different temporal dynamics

Since alpha and beta rhythms are anatomically and functionally separated at the cortical level, we asked whether the neuronal ensembles supporting the two sensorimotor rhythms provide different contributions to imagined movements. We considered the temporal dynamics of power changes in alpha- and beta-band rhythms, aggregated across the relevant local maxima. These temporal dynamics were highly correlated ($r = 0.7 \pm 0.1$, M ± SEM, p<0.002) and both alpha- and beta-band power was more strongly attenuated for the hemisphere contralateral to the arm used in the imagined movement, see *Figure 2E*. Yet it can be seen from the same graph that alpha-band power increases in the (postcentral) cortex ipsilateral to the arm used for imagery, as compared to baseline levels (+34% between 910 and 1220 ms, p<0.05; alpha-band power also decreased by 26% and 32% in the contralateral cortex between 170 and 850 ms and between 1230 and 2000 ms, respectively). In contrast, beta-band power decreases further in the (pre- and postcentral) contralateral cortex (−21% between 150 and 760 ms vs. −13% in the ipsilateral cortex between −180 and 580 ms; there was another statistically significant change of −21% from baseline in the contralateral cortex between 1450 and 2000 ms). These differential power changes are robust on the single-trial level and, as seen in *Figure 2—figure supplement 2*, represented modulations of sustained rhythmic activity (*Jones, 2016*; *Little et al., 2018*).

The temporal dynamics of these power changes are highly consistent with previous observations obtained from non-invasive electrophysiological recordings over sensorimotor cortex during performance of the same task (cf. Figure 3 in *Brinkman et al., 2014*). In that magnetoencephalography (MEG) study, it was observed that as selection demands increased (when cylinder orientations afforded both over- and underhand grasping), alpha-band power increased in the sensorimotor cortex ipsilateral to the arm used for motor imagery, whereas beta-band power concurrently decreased in the contralateral sensorimotor cortex. We examined the alpha- and beta-band local maxima for similar effects, although the patients recorded in this ECoG study performed a substantially lower number of trials than the healthy participants of the MEG study (120 vs. 480, respectively). We defined high demand trials as trials involving cylinders oriented around the switch points estimated from each hand's response curve (range: three orientation bins per switch point, that is −24° to +24°). We compared alpha- and beta-band temporal dynamics on high demand trials with those on low demand trials, defined as trials with cylinder orientations orthogonal to the switch points and covering an equivalent range. It can be seen from *Figure 2—figure supplement 3* that the direction of the effects is consistent with the previous MEG observations. There was a statistically significant

decrease in contralateral beta rhythmic activity with increasing demand. However, the increase in ipsilateral alpha rhythmic activity did not pass the statistical threshold. Concerns regarding the limited number of trials refrained us from using the effects of task demand for further analyses.

## Alpha- and beta-band rhythms arise from spatiotemporally unrelated neuronal ensembles

Since the temporal dynamics of alpha and beta rhythms aggregated across local maxima is functionally divergent, we asked whether that dissociation persists at more fine-grained levels of analysis across ECoG electrodes and trial-by-trial sensorimotor demands. First, we considered the temporal and spatial correlations between alpha- and beta-band power both between their respective local maxima (*Figure 3A*) and across the same functionally demarcated sensorimotor cortex (*Figure 3B, C*). It can be seen from the leftmost bars in these figures that alpha- and beta-band rhythms were temporally as well as spatially uncorrelated (all $BF_{01}$ >1.56 in favor of the null hypothesis of no correlation). This finding is a merit of the current procedure separating alpha and beta rhythmic activity from concurrent $1/f$ modulations in the power spectrum, as power in the two frequency bands was correlated when this shared variance was not accounted for (*Figure 3—figure supplement 1*). Second, we considered the representational similarity of the temporal and spatial activity patterns evoked during movement imagery in the alpha- and beta-bands (*Kriegeskorte et al., 2008*). Instead

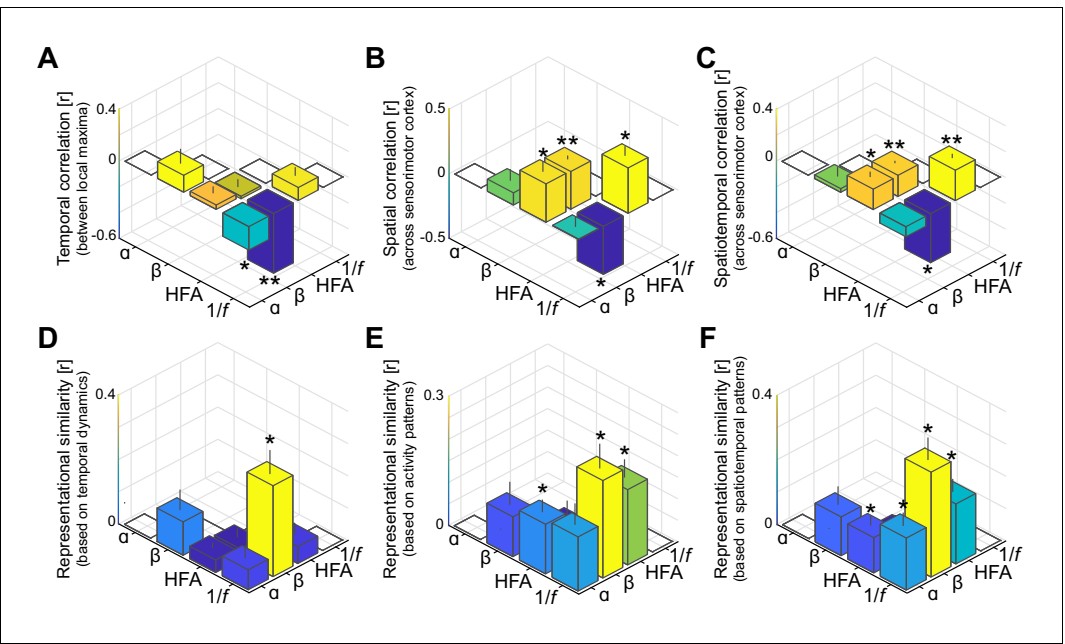

**Figure 3.** Spatiotemporal dissociation of sensorimotor alpha and beta. (A - C) Temporal, spatial, and spatiotemporal correlations between alpha, beta, high-frequency activity (HFA), and the $1/f$ slope. Alpha and beta rhythms were weakly correlated in time and space during movement. Both alpha and beta showed a positive relationship with high-frequency activity, yet only beta-band power closely tracked changes in the $1/f$ slope across sensorimotor cortex (B and C). *: p<0.05; **: p<0.001. (D - F) Alpha and beta rhythms showed weak similarity in sensitivity to sensorimotor demands across different movements. Echoing the correlations shown in panels A to C, beta is largely sensitive to the same trial-by-trial demands as the $1/f$ slope, for both sensorimotor demands contained by temporal dynamics (D) and activity patterns (E and F).

DOI: https://doi.org/10.7554/eLife.48065.007

The following figure supplements are available for figure 3:

**Figure supplement 1.** As in *Figure 3A–C*, but without accounting for shared variance in alpha- and beta-band frequency bands originating from concurrent $1/f$ modulations in the power-spectrum.
DOI: https://doi.org/10.7554/eLife.48065.008

**Figure supplement 2.** As in *Figure 3A–C*, but with high-frequency activity and the $1/f$ slope index based on the rhythmic component rather than on the arrhythmic component of the power-spectrum.
DOI: https://doi.org/10.7554/eLife.48065.009

of calculating direct correlations between the temporal dynamics or the spatial distribution of alpha- and beta-band power as above, this second-order correlation analysis quantifies the similarity in sensitivity to sensorimotor demands across trials, independently from the frequency-specific neural patterns evoked within a trial. Alpha- and beta-band rhythms showed weak resemblances in sensitivity to trial-by-trial demands, for both sensorimotor demands contained by temporal dynamics and activity patterns (*Figure 3D–F*, $BF_{01}s$ of 1.06, 1.01, and 0.73, respectively). These relations between alpha- and beta-band effects indicate that the neuronal ensembles producing these two sensorimotor rhythms have no substantial spatiotemporal correspondences, neither within trials nor across trials.

## Alpha- and beta-band rhythms have different influence on local excitability

The previous sections provide evidence for the notion that the neuronal ensembles generating alpha- and beta-band rhythms have different spatiotemporal characteristics during motor imagery, as well as different peripheral consequences following electrical stimulation. These observations confirm and qualify the findings of previous ECoG and SEEG reports on differences between alpha- and beta-band rhythms over the sensorimotor cortex (*Brovelli et al., 2004*; *Crone et al., 1998*; *Jasper and Penfield, 1949*; *Saleh et al., 2010*; *Szurhaj et al., 2003*; *Toro et al., 1994*; *Vansteensel et al., 2013*). Those clear differences between alpha- and beta-band rhythms raise the issue of understanding the functional consequences of those differences on the excitability of neuronal populations in the sensorimotor cortex during movement imagery. We indexed those consequences through spectral markers of local population-level activity (arrhythmic high-frequency activity between 60 and 120 Hz; *Manning et al., 2009*; *Miller et al., 2009*; *Ray and Maunsell, 2011*) and of local excitation/inhibition balance (steepness of the power-spectral 1/*f* slope, estimated between 30 and 50 Hz; *Gao et al., 2017*). High-frequency activity showed spatial and temporal correspondences with both alpha- and beta-band rhythmic activity during movement imagery (*Figure 3B,C*). This is also seen in the spatial distribution of local maxima in high-frequency activity (green electrodes in *Figure 2D*), which were localized to both sides of the central sulcus and involved neuronal ensembles producing alpha- or beta-band rhythmic activity (14/22: four producing alpha, four producing beta, six producing both alpha and beta, and eight with no overlap). However, the lack of clear effector-specificity (*Figure 2E*) limits the functional relevance of this index.

Unlike high-frequency activity, the 1/*f* slope index showed clear functional specificity. This index was sensitive to the laterality of the effector involved in the motor imagery task (*Figure 2E*). This index was also spatially specific, with a focal reduction of excitation/inhibition ratio (i.e., steepest 1/*f* slopes, indicating stronger local inhibition) at electrodes placed over the central sulcus yielding predominantly somatomotor rather than somatosensory responses following electrical stimulation ($\chi^2(27)=10.3$, p<0.002; orange electrodes in *Figure 2C,D*). The spatial specificity of the 1/*f* slope index is further supported by a direct comparison with the spatial distribution of high-frequency activity: despite superficially similar distributions across the central sulcus (*Figure 2D*), only 3 out of 47 combined unique electrodes were both local maxima for high-frequency activity and local inhibition as indexed by the 1/*f* slope. One of the main findings of this study is that the 1/*f* slope index had a differential relationship with the two sensorimotor rhythms. *Figure 3A–C* illustrates the reciprocal changes observed between beta-band activity and the 1/*f* slope during task performance. Namely, stronger reductions in beta-band power correlated with stronger increases in local excitability across sensorimotor cortex. Furthermore, electrodes with local maxima in beta-band activity and local inhibition were similarly distributed across the central sulcus, with a 59% (10/17) spatial correspondence. Given that both beta-band and 1/*f* slope indexes were similarly responsive to the laterality of the effector involved in the motor imagery task (*Figure 2E*), the spatiotemporal correspondence between beta-band rhythm and 1/*f* slope indicates that the stronger beta-band power reduction in the somatomotor cortex contralateral to the selected arm is associated with a relative disinhibition of somatomotor neuronal populations. This inference is supported and generalized by the representational similarity analyses of the temporal and spatial relations between those two spectral indexes evoked during movement imagery (*Figure 3D–F*). These analyses indicate that there is a robust spatiotemporal similarity across different imagined movements between beta-band power and 1/*f* slope, over and above the within-trial correlations captured in *Figure 3A–C*.

In contrast, the 1/$f$ slope index had a different relationship with alpha-band responses to task demands. The putative index of excitation/inhibition balance was not spatially related to the alpha-band response (*Figure 3B,C*), with a 25% correspondence (5/20) between electrodes with local maxima in alpha-band activity and local inhibition. However, there was a significant temporal anti-correlation between local maxima of alpha-band power and 1/$f$ slope (*Figure 3A*). This observation suggests that the stronger alpha-band power evoked in the somatosensory cortex ipsilateral to the selected arm (*Figure 2E*) is associated with a relative but spatially unspecific inhibition of the sensorimotor cortex. This inference is partially supported by the representational similarity analyses (*Figure 3D–F*). Although the trial-by-trial variation in spatiotemporal patterns of alpha-band power and 1/$f$ slope is significantly related (*Figure 3F*), there are no clear similarities between those two spectral indexes when only temporal or spatial profiles are considered (*Figure 3D,E*).

## Alpha- and beta-band rhythms propagate independently across sensorimotor cortex

The differential relation of alpha- and beta-band rhythms to (dis)inhibition of the sensorimotor cortex raises the issue of understanding whether that (dis)inhibition is propagated in a consistent spatiotemporal pattern. This possibility is functionally relevant: It has been suggested that there are consistent phase relationships among rhythmic cortical signals, organized in sparse traveling waves that could facilitate sequences of activation in proximal-to-distal muscle representations in preparation for reaching behavior (*Ermentrout and Kleinfeld, 2001*; *Muller et al., 2018*). We explored this possibility by assessing the traveling wave characteristics of ECoG signals filtered at individual alpha- and beta-band frequencies and examining the functional relationship of those traveling waves with neuronal ensembles generating alpha and beta rhythms.

Visual inspection of single-trial filtered activity indicated that the phase of alpha- and beta-band signals varied systematically across the electrode array during motor imagery (*Figure 4A*). To quantitatively verify that rhythmic activity spatially progressed as traveling waves across sensorimotor cortex, we estimated spatial gradients of instantaneous rhythm phase computed using the Hilbert transform at each electrode across the recording array. These spatial gradients represent distance-weighted phase shifts between cortical signals at neighboring recording electrodes, where positive phase shifts correspond to signals that have covered a greater distance along the unit circle and thus lead the oscillation. To test whether the spatial gradients behaved like propagating waves at the single-trial level, we computed the phase-gradient directionality (PGD), a measure of the degree of phase-gradient alignment across an electrode array (*Rubino et al., 2006*). As seen through the small cone-shaped arrows positioned over each corresponding grid-electrode in *Figure 4A*, both alpha and beta phase gradients exhibited a higher degree of alignment across sensorimotor cortex than expected by chance (mean alpha PGD = 0.37, mean beta PGD = 0.35, p<0.001 in each patient for both alpha and beta, estimated from shuffled data). The traveling waves moved in a consistent direction across trials and over trial-time (circular histograms in *Figure 4A*; Rayleigh test of uniformity, p<10$^{-18}$ in 5 out of 6 patients for alpha, p<10$^{-91}$ in each patient for beta). Across participants, mean propagation speeds of the sensorimotor waves ranged between 5 and 9 cm/s for alpha and between 11 and 21 cm/s for beta (*Figure 4B*), consistent with previous reports of traveling beta waves in motor cortex (*Rubino et al., 2006*) and in the lower range of traveling alpha waves observed in posterior cortex (*Bahramisharif et al., 2013*; *Halgren et al., 2017*; *Zhang et al., 2018*). These observations corroborate and extend previous studies by showing that both alpha- and beta-band rhythms are organized in waves traveling across the sensorimotor cortex (*Halgren et al., 2017*; *Takahashi et al., 2015*; *Zhang et al., 2018*).

A novel finding of this study is that alpha and beta traveling waves propagate independently across sensorimotor cortex, as indicated by the distribution of propagation directions in individual participants (*Figure 4A*, *Videos 1* and *2*) and by the mean probability distribution over participants (*Figure 4C*; mean Kullback-Leibler divergence = 0.10, p<0.001 in each patient, estimated from shuffled data). Alpha waves propagated in a caudo-rostral direction, while beta waves advanced in a rostro-caudal direction (*Figure 4C*, *Figure 4—figure supplement 1*). This analysis also revealed that electrodes sampling alpha- or beta-band rhythms with larger amplitudes were not sources or sinks of the alpha- or beta-traveling waves: Previously identified local maxima in alpha- and beta-band activity did not have a systematic phase advantage or delay in relation to other electrodes across the sensorimotor cortex (*Figure 4A*). Nevertheless, traveling-wave-like activity at these cortical sites

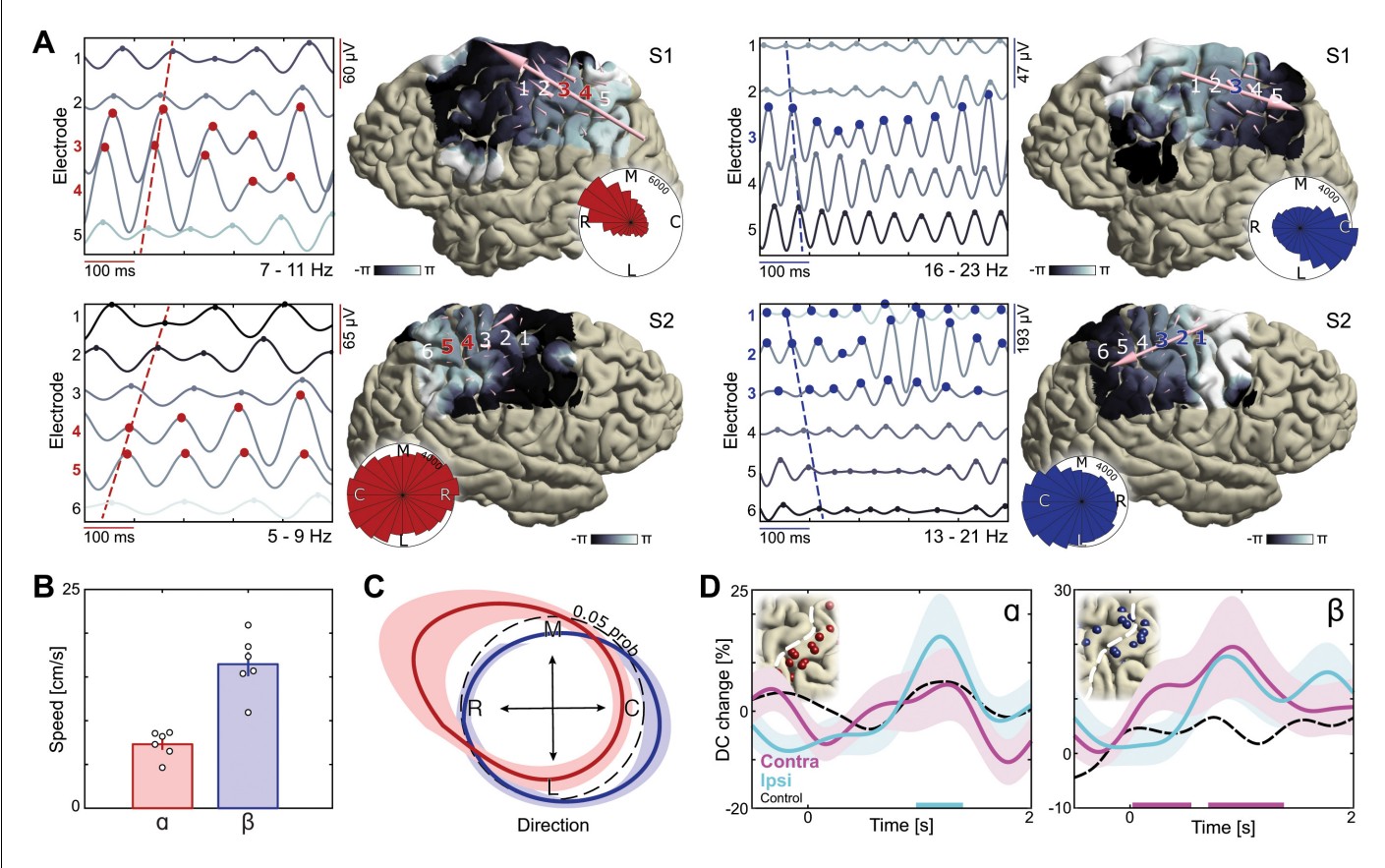

**Figure 4.** Dissociation of sensorimotor alpha and beta traveling waves. (**A**) Propagation of alpha and beta rhythmic activity during imagined movement in two representative individuals. Example cortical signals are of the same data segment in each participant but filtered at individual alpha and beta frequencies. Red and blue markers indicate electrodes previously identified as alpha- and beta-band local maxima, respectively. Cortical phase maps indicate the average phase at each cortical site relative to a central sensorimotor reference electrode. Small cone-shaped arrows indicate the mean propagation direction at each stimulation-positive electrode, with arrow size weighted by the local phase gradient magnitude. Large arrows indicate the mean propagation direction across sensorimotor cortex, with arrow size weighted by the alignment of sensorimotor gradients (phase gradient directionality, PGD). (**B**) Mean propagation speeds of traveling alpha and beta waves over participants. (**C**) Mean probability distribution of traveling wave direction over participants. Alpha rhythm propagation is maximal in a caudo-rostral direction (red distribution), while beta rhythms predominantly moved in a rostro-caudal direction (blue distribution). Dashed black circle represents a uniform distribution of propagation directions, for reference. (**D**) Alpha traveling waves propagated more consistently through alpha-band local maxima during imagined movement of the ipsilateral arm (directional consistency, DC). In contrast, beta waves traveled more consistently through beta-band local maxima during imagined movement of the contralateral arm. Colored bars along the x-axes indicate time intervals of statistically significant DC changes from baseline levels for the effector involved in the imagined movement.

DOI: https://doi.org/10.7554/eLife.48065.010

The following figure supplements are available for figure 4:

**Figure supplement 1.** Cross-correlation functions of alpha and beta rhythmic activity at rostro-caudal electrode pairs on the sensorimotor cortex of two representative individuals.

DOI: https://doi.org/10.7554/eLife.48065.011

**Figure supplement 2.** As in *Figure 4D*, but with directional consistency of wave propagation for high demand trials (solid lines, cylinder orientations that afforded both overhand and underhand grasping) and for low demand trials (dashed lines, cylinder orientations that afforded grasping in a single manner only).

DOI: https://doi.org/10.7554/eLife.48065.012

was task-relevant, as indicated by an increase in directional consistency (DC) of those waves during movement imagery. Directional consistency measures the degree of consistency across trials in the phase-gradient direction (*Zhang et al., 2018*). As seen in *Figure 4D*, alpha rhythms propagated in a more consistent direction during imagined movement of the ipsilateral arm, while the propagation

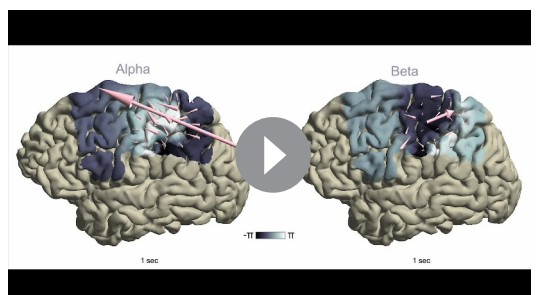

**Video 1.** Time-lapse video of concurrent traveling alpha and beta waves in participant S1 during movement imagery. Cortical phase maps indicate the average phase at each cortical site relative to a central sensorimotor reference electrode. Small cone-shaped arrows indicate the mean propagation direction at each stimulation-positive electrode, with arrow size weighted by the local phase gradient magnitude. Large arrows indicate the mean propagation direction across sensorimotor cortex, with arrow size weighted by the alignment of sensorimotor gradients (phase gradient directionality, PGD). Time is in seconds after cylinder appearance.

DOI: https://doi.org/10.7554/eLife.48065.013

direction of beta rhythms became more consistent during imagined movement of the contralateral arm, as compared to baseline levels (see *Figure 4—figure supplement 2* for the effects of task demand). Together, these observations indicate that the broader spatiotemporal context in which rhythmic cortical signals are embedded constitute an important component of the movement selection demands evoked by motor imagery, and that this spatiotemporal organization differs for alpha and beta rhythms.

## Discussion

This ECoG study qualifies the spatiotemporal dynamics of alpha- and beta-band rhythms and their effects on the local excitability of sensorimotor neuronal ensembles during movement imagery. Rhythmic signals in the alpha- and beta-band were prominent in the patients' sensorimotor cortex, sustained across each trial, motorically relevant, and organized in spatially consistent waves of phase relationships traveling along opposite directions. In line with previous reports (*Brinkman et al., 2014*; *Crone et al., 1998*; *de Lange et al., 2008*; *Miller et al., 2010*), this study shows that the power envelopes of those two rhythms differentiated between imagined movements involving the contralateral or the ipsilateral arm. This study also confirms historical accounts by showing that alpha- and beta-band rhythms arise from anatomically and functionally distinct neuronal ensembles (*Berger, 1938*; *Jasper and Penfield, 1949*; *Salmelin and Hari, 1994*). Local maxima of alpha-band power were distributed on the postcentral gyrus, and electrical stimulation of those electrodes yielded somatosensory sensations of the upper limb. Sensorimotor beta was strongest at electrodes placed over the central sulcus, with electrical stimulation yielding both movements and somatosensory sensations. This study provides a novel piece of empirical evidence showing that sensorimotor alpha and beta rhythms have different neurophysiological properties, (dis)inhibiting dissociable sensorimotor neuronal ensembles. Namely, beta rhythmic activity closely tracked task-related modulations of the 1/*f* slope of the power-spectrum, an index of excitation/inhibition balance (*Gao et al., 2017*). The relation between beta and 1/*f* slope held across the spatial extent of the sensorimotor cortex, and within trials as well as across trials. When the 1/*f* slope transiently increased in somatomotor cortex during movement imagery, indicating a shift in balance from relative inhibition to excitation, beta rhythmic activity showed a focal reduction in signal strength. These findings suggest that imagery-related reduction in beta-band power, predominant over the somatomotor cortex contralateral to the selected arm, is associated with a relative disinhibition of somatomotor neuronal populations. This beta-band movement-related disinhibition was embedded within traveling waves moving along a rostro-caudal direction across the fronto-parietal cortex. There was also a relative increase in alpha-band power in the somatosensory cortex ipsilateral to the selected arm, an effect that was associated with a spatially unspecific inhibition of the sensorimotor cortex. This alpha-band inhibition was embedded within traveling waves along a caudo-rostral direction across the parieto-frontal cortex. We draw two main conclusions from these human neurophysiological observations. First, the evidence points to the relevance of both disinhibition and inhibition mechanisms for precise spatiotemporal coordination of movement-related neuronal populations. Second, the evidence points to the dramatically different neurophysiological properties of sensorimotor alpha and beta rhythms, questioning the practice of aggregating those rhythms when studying cerebral function.

These findings emphasize how increased excitability of the sensorimotor cortex goes hand in hand with increased (and spatially widespread) inhibition. Speculatively, the spatiotemporal profile

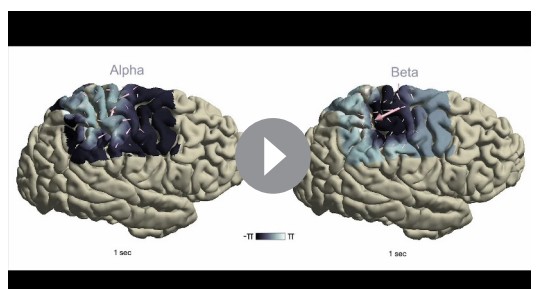

**Video 2.** As in *Video 1*, for participant S2.
DOI: https://doi.org/10.7554/eLife.48065.014

of increased excitability observed in the contra-lateral sensorimotor cortex might support the coordination of multiple sensorimotor cortical ensembles toward a movement-effective neural subspace (*Elsayed et al., 2016*; *Shenoy et al., 2013*), possibly implemented as dynamic modulations in direction- and frequency-dependent spatial arrangements of neuron receptor fields (*Heitmann et al., 2013*). Accordingly, beta waves in the motor cortex carry most movement-related information during the preparatory phase of a movement (*Rubino et al., 2006*). The spatially unspecific inhibition of the ipsilateral sensorimotor cortex suggests that coordinating complex movements also requires suppression of task-irrelevant movements and in particular inhibition of their somatosensory correlates. It seems unlikely that this alpha inhibitory effect was driven by somatosensory attention to the hand used during imagery since there were no lateralized power changes in the prestimulus baseline period, during which participants knew which hand they would use.

## Interpretational issues

Previous micro-ECoG studies in non-human primates have shown systematic phase relationships between motor cortical signals less than a millimeter apart (*Rubino et al., 2006*; *Takahashi et al., 2015*). Here, we add to those findings by showing that alpha- and beta-band traveling waves propagate across the human sensorimotor cortex, independently. High-density laminar recordings of alpha and beta rhythmic activity might be able to test whether those rhythms propagate through different cortical layers (*van Kerkoerle et al., 2014*). Another possibility is that different cortico-thalamo-cortical and cortico-striatal-thalamo-cortical circuits lead to different alpha and beta traveling waves across the sensorimotor cortex (*Bastos et al., 2014*; *Schreckenberger et al., 2004*; *West et al., 2018*). The latter possibility could accommodate the observation that sources/sinks of the traveling waves were independent from electrodes sampling rhythms with larger amplitudes, and that there were no obvious phase-shifts between neighboring electrodes spanning a cortical fold. Large-scale corticothalamic recordings of alpha and beta waves might be able to define the precise mechanisms supporting those traveling waves over human sensorimotor cortex (*Halgren et al., 2017*).

Alpha- and beta-band rhythms are embedded within (but physiologically different from) arrhythmic broadband $1/f$ components of the signal, and their spectral distributions differ between individuals (a case in point is participant S7 lacking a rhythmic component in the alpha frequency range). Supplementary analyses indicate that ignoring those facts, as standard analytical pipelines do, led to strong but spurious correlation between alpha and beta power envelopes. Furthermore, the spatial separation between alpha- and beta-band cortical sources might prove too subtle for many non-invasive electrophysiological recordings (*Brinkman et al., 2014*; *Fransen et al., 2016*). These considerations might help to understand why the two sensorimotor rhythms are often aggregated into the same (mu-) rhythm category (*Cuevas et al., 2014*; *Hari, 2006*). Having shown that alpha and beta rhythms are anatomically and functionally distinct phenomena, it becomes relevant to know whether the two rhythms can also be systematically differentiated in other frontal brain regions (*Bastos et al., 2018*; *Johnston et al., 2019*).

The neural effects measured in this study did not always have a clear behavioral correlate beyond effector-specificity. By contrast, a previous MEG study using the same task showed clear and opposite relationships between alpha/beta effects and imagery demands (*Brinkman et al., 2014*). This ECoG study involved individuals undergoing presurgical monitoring, a relatively rare clinical procedure with limited opportunities for experimentally controlled observations. The direction of neural effects related to task demand was consistent with the previous observations, yet statistical power might have been too low for effects based on a subset of trials. Accordingly, this study refrained from in-depth explorations of the effects of task demand, focusing on the ECoG recordings' anatomical precision and signal-to-noise ratio to provide a clear neurophysiological characterization and differentiation of alpha and beta rhythmic activity in the human sensorimotor system.

## Conclusions

The current findings indicate that alpha- and beta-band rhythms, besides having different anatomical distributions and traveling along opposite directions across the sensorimotor cortex, have different effects on cortical excitability. Increased alpha rhythmic activity in the somatosensory cortex ipsilateral to the arm selected for motor imagery is associated with spatially-unspecific cortical inhibition, whereas a reduction in beta rhythmic activity over contralateral motor cortex is associated with a spatially-focal shift in excitation/inhibition balance toward excitation. These findings increase our understanding of how cortical rhythms can mechanistically support the precise spatiotemporal organization of neuronal ensembles necessary for coordinating complex movements in humans.

# Materials and methods

## Key resources table

| Reagent type (species) or resource | Designation | Source or reference | Identifiers | Additional information |
|---|---|---|---|---|
| Software, algorithm | FieldTrip | FieldTrip | *Stolk et al., 2018* | Integrated analysis of human intracranial data |
| Software, algorithm | FreeSurfer | FreeSurfer | *Dale et al., 1999* | Cortical surface extraction |

## Participants

Eleven participants (7 males, 14–45 y of age) were implanted subdurally with grid and strip electrode arrays on the cortical surface to localize the seizure onset zone for subsequent surgical resection (*Figure 1A*). The electrode arrays (10 mm inter-electrode spacing, 2.3 mm exposed diameter; Ad-Tech, Racine, USA) were placed at the University Medical Center Utrecht, The Netherlands, on either right or left (eight cases) hemisphere. The number and anatomical location of the electrodes varied across participants, depending on the clinical considerations specific to each case (mean number of electrodes ± SEM: 81.3 ± 11.2). The sample size was determined by the availability of participants with (partial) electrode coverage of the central sulcus during the funding period of the project (four years). All participants had normal hearing and normal vision, and gave informed consent according to institutional guidelines of the local ethics committee (Medical Ethical Committee of the University Medical Center Utrecht), in accordance with the declaration of Helsinki. No seizures occurred during task administration. Two participants had difficulties adhering to the task instructions and frequently confused left- and right-hand conditions of the study. One of these participants had cavernous malformations in temporoparietal and frontal cortex. The other participant had experienced medical complications prior to task performance, leaving nine participants for analysis of the behavioral data. Two participants had no electrode coverage of upper-limb sensorimotor areas as indicated by electrocortical stimulation, leaving seven participants for analysis of the neural data.

## Movement imagery task

Participants were positioned in a semi-recumbent position in their hospital bed and performed up to three sessions of a movement imagery task (mean number of sessions ± SEM: 2 ± 0.2). In this task, participants imagined grasping the middle-third of a black-white cylinder with either their left or right hand (*Figure 1B*). The cylinder, tilted according to 1 of 15 possible orientations (24° apart, presented pseudo-randomly, size 17.5 × 3.5 cm), was presented on a gray background at the center of the computer screen that was placed within reaching distance in front of the participant. The duration for which the cylinder stayed on the screen was adjusted for each participant (2–5 s) such that they could comfortably perform the task at a pace that suited their current physical and mental state. Next, a response screen appeared where the participants indicated whether their thumb was on the black or the white part of the cylinder at the end of the imagined movement. The response screen consisted of two squares on the horizontal plane (one black and one white), where participants indicated 'black' or 'white' by pressing the corresponding button (left or right button) using the left or right thumb on a button box that they held with both hands. The order of the squares (black - left,

white - right, or vice versa) was pseudo-random across trials to prevent the preparation of a response during the simulation of the grasping movements. After the response, a fixation cross appeared on the screen for 3 to 4 s (drawn randomly from a uniform distribution), after which the next trial started (intertrial interval). A single session consisted of 60 trials (10 min). The hand used to imagine the movement alternated every ten trials, prompted by a visual cue. The task exploited the fact that certain cylinder orientations afforded both overhand and underhand grasping, whereas other orientations afforded grasping in a single manner only, due to biomechanical constraints of the hand (*Figure 1C*). This task manipulation provided a test of participants' imagery performance as to whether their preferred manner for grasping the cylinder (thumb on black or white part) was modulated by biomechanical constraints, varying as a function of cylinder orientation and differing for the left and right hand.

Eight out of nine participants whose behavioral data are reported (5 out of 6 participants whose neural data are reported), completed a control task that used the same visual input and response contingencies, but where no imagery was required. In the control task, participants reported which side of the black-white cylinder was larger. That is, the surface areas differed slightly across trials, for example 54% black and 46% white, or vice versa. This allowed controlling for neural changes unrelated to the movement imagery process, such as those evoked by visual input during task performance.

## ECoG acquisition and analysis

Electrophysiological data were acquired using the 128-channel Micromed recording system (Treviso, Italy, 22 bits), analog-filtered between 0.15 and 134.4 Hz, and digitally sampled at 512 Hz. During the recordings, participants were closely monitored for overt movements or distracting events. Epochs were these occurred were excluded from the analysis (6 ± 2% of the total amount of trials). Anatomical images were acquired using preoperative T1-weighted Magnetic Resonance Imaging (MRI, Philips 3T Achieva; Best, The Netherlands) and post-implantation Computerized Tomography (CT, Philips Tomoscan SR7000).

Data were analyzed using the open-source FieldTrip toolbox (*Oostenveld et al., 2011*), performing an integrated analysis of anatomical and electrophysiological human intracranial data. The procedure for the precise anatomical registration of the electrophysiological signal in each patient is described in detail elsewhere (*Stolk et al., 2018*). In brief, electrode locations in relation to the brain's anatomy and the electrophysiological signal were obtained through identification of the electrodes in a post-implantation CT fused with the preoperative MRI. To correct for any displacement following implantation, the electrodes were projected to individually rendered neocortical surfaces along the local norm vector of the electrode grid (*Hermes et al., 2010*). We used FreeSurfer to extract anatomically realistic neocortical surfaces from each participant's MRI (*Dale et al., 1999*). FreeSurfer also allows registering the surfaces to a template brain on the basis of their cortical gyrification patterns (*Greve et al., 2013*). Using these surface registrations, we linked the electrodes from all participants to their template homologs, preserving the spatial relationship between cortical folding and electrode positions in each participant. This allowed for anatomically accurate comparison of local maxima in neural activity across participants.

The electrophysiological signals were visually inspected to ensure that they were free of epileptic activity or other artifacts (2 ± 2% of the total amount of trials excluded). Next, the data were digitally filtered (1–200 Hz bandpass, Butterworth, zero-phase forward and reverse), removed from power line noise components (50 Hz and harmonic band stop), and re-referenced to use an average reference. This produced cortical signals removed from activity common to all channels. We focused the analysis on the trial epochs during which the participants imagined a movement, preceded by the appearance of the black-white cylinder. Using time-resolved Fourier analysis, we calculated spectral power with 1000 ms rolling Hanning-tapered windows at 50 ms increments. This produced time-frequency estimates up to 200 Hz with a 1 Hz spectral and a 20 Hz temporal resolution. Inter-session offsets in absolute spectral power were compensated for using linear regression analysis considering mean power across all time-frequency estimates in a session. For temporal dynamics analysis, the spectral data were expressed as percentage changes from bootstrapped spectral power during a pre-cylinder baseline interval (−750 to −500 ms to cylinder onset) and resampled to identical duration across participants (2 s, after anti-aliasing). Differences in spectral power between the left- and right-hand conditions were evaluated using nonparametric cluster-based permutation statistics (two-

sided dependent samples *t*-tests, p<0.05, 10,000 randomizations; *Maris and Oostenveld, 2007*), considering electrodes containing local maxima in neural activity as the unit of observation.

## Spectral features extraction from sensorimotor cortex

Alpha and beta spectral and anatomical distributions were defined on a participant-by-participant basis, using a four-step procedure. First, electrodes covering cortical regions supporting sensorimotor components of movement were identified using Electrocortical Stimulation Mapping (ESM, Micromed IRES 600CH), a standard clinical practice involving the pairwise electrical stimulation of adjacent cortical electrodes (typically at 50 Hz for 1–2 s, with a 0.2–0.5 ms pulse duration and 1–4 mA intensity). Intensity of the stimulation was individually tailored, maximizing effect size while minimizing the occurrence of after-discharges. For each participant, we selected electrodes that were part of a stimulation electrode pair yielding motor or somatosensory responses of the upper limb contralateral to the cortical grid (twitches, movements, tingling of either fingers, hand, wrist, arm or shoulder).

Second, we used irregular-resampling auto-spectral analysis (IRASA, *Wen and Liu, 2016*) of the signal recorded at the stimulation-positive electrodes, allowing distinguishing rhythmic activity from concurrent power-spectral 1/*f* modulations. This technique virtually compresses and expands the time-domain data with a set of non-integer resampling factors prior to Fourier-based spectral decomposition, redistributing rhythmic components in the power-spectrum while leaving the arrhythmic 1/*f* distribution intact. Taking the median of the resulting auto-spectral distributions extracts the power-spectral 1/*f* component, and the subsequent removal of the 1/*f* component from the original power-spectrum offers a power-spectral estimate of rhythmic content in the recorded signal. It should be noted that the extracted spectral components no longer contain phase information and that their estimated magnitudes are susceptible to any phase relationships between the two components, as indicated by Equation 9 in the original paper (cf. two opposite-phase oscillations canceling out one another in the summed signal). As a consequence, power in the rhythmic component is negative at frequencies where the arrhythmic 1/*f* component exceeds power of the original power-spectrum. In cases where this happened (never at spectral peaks), we set power to zero to accommodate spectral curve fitting with exponential models in the next step.

Third, mean and full-width at half-maximum of alpha and beta spectral distributions were defined for each participant using a two-term or three-term Gaussian model, depending on the presence of a third low-frequency phenomenon in the rhythmic component of the power-spectrum (<5 Hz in two participants, see power-spectra in *Figure 2—figure supplement 1*). This adaptive approach (*Source code 1*) avoids having to rely on canonical frequency bands that due in part to their narrowness may not accurately capture the neural phenomena of interest in each individual (*Haegens et al., 2014*; *Szurhaj et al., 2003*). On average, alpha and beta rhythmic activity were centered on 7.4 ± 0.7 and 16.9 ± 1.1 Hz, respectively. High-frequency neural activity was defined as activity within a broad 60–120 Hz range (*Lachaux et al., 2012*). Because of its hypothesized relationship with non-oscillatory population-level firing rate (*Manning et al., 2009*; *Miller et al., 2009*; *Ray and Maunsell, 2011*), we estimated high-frequency activity using the arrhythmic 1/*f* component obtained above (see also *Figure 3—figure supplement 2* for an empirical argument). We additionally considered the slope of the arrhythmic 1/*f* component, in log-log space. Computational modeling and local field potential recordings from rat hippocampus suggest that the slope between 30 and 50 Hz is a power-spectral correlate of synaptic excitation/inhibition balance, such that a steeper slope corresponds to greater inhibition in a neuronal ensemble measured by the recording electrode. Notably, electrocorticography recordings in the non-human primate brain indicate that the 1/*f* slope closely tracks the increase of inhibition induced by propofol across space and time (*Gao et al., 2017*). Furthermore, recent intracranial recordings in humans find that the slope between 30 and 50 Hz best predicts the depth of sleep and anesthesia, more so than slow oscillatory power (*Lendner et al., 2019*). We here assessed this measure's potential for capturing movement initiation and suppression in human sensorimotor cortex. Linear fits were used to estimate the steepness of the slope in the 30–50 Hz range (mean $R^2$ across all slope fits in each individual = 0.95 ± 0.00).

Fourth, for a fine-grained anatomical characterization, we localized all four sensorimotor neuronal phenomena (alpha and beta rhythmic activity, high-frequency arrhythmic activity, and the 1/*f* slope) by selecting electrodes that exceeded the upper limit of the 99% confidence interval for absolute

spectral power in the respective frequency band across all stimulation-positive electrodes defined by the first step (for the 1/f slope we used the lower limit of the confidence interval). This analysis yielded 4 ± 1.2 alpha, 3.4 ± 0.8 beta, 4.4 ± 0.7 high-frequency, and 5.6 ± 1.4 1/f slope local maxima in sensorimotor cortex for participants S1 - 5. Due to limited sensorimotor coverage, the number of electrodes could not be narrowed down for participant S6, and all four stimulation-positive electrodes were considered for further analysis involving temporal dynamics. Participant S7 lacked a rhythmic power-spectral component in the alpha frequency range and was excluded from further analysis.

We used chi-squared tests of electrode anatomical location and electrical stimulation response type to assess differential basic sensorimotor properties of alpha and beta rhythms. Anatomical location was defined as the electrode's spatial relationship to the central sulcus (pre vs. postcentral sulcus), and response type as the sensorimotor nature of the evoked response following electrical stimulation (motor response vs. somatosensory sensation).

## Spatiotemporal relations between spectral features

To assess whether sensorimotor alpha, beta, high-frequency activity, and the 1/f slope shared features during task performance, we performed a correlation analysis of their activity patterns across time, space, as well as time and space combined. First, within-trial correlations of activity dynamics between −750 and 2000 ms (relative to the onset of the visual stimulus) quantified the temporal similarity between the four spectral features. These temporal correlations considered, for each participant, mean activity across local maxima of each spectral feature (as identified with the procedure described above). Each pair of spectral features produced a single correlation value per trial. Second, a similar procedure was used to assess whether those spectral features involved spatially overlapping or distinct neuronal ensembles across sensorimotor cortex. We considered within-trial correlations of cortical activity patterns across stimulation-positive electrodes. In contrast to temporal correlation, spatial correlation considered the mean activity per electrode within a trial (converted into a vector), from visual stimulus presentation onset until the end of the movement imagery interval (0 to 2000 ms). A third correlation analysis quantified the similarity of spatiotemporal activity patterns across all stimulation-positive electrodes during a trial (−750 to 2000 ms). Group-level analysis considered the average correlation in each participant, where the reliability of these correlations across the sample population was assessed using one-sample t-tests. We report Bayes Factors (BF$_{01}$) for statistical tests evaluating evidence in favor of the null hypothesis. Bayes Factors express the relative likelihood of the data under the models at hand and were calculated using the JASP statistical software package (JASP Team, https://jasp-stats.org/).

To assess whether the different neural phenomena were sensitive to the same sensorimotor demands across individual movements, we performed representational similarity analysis on temporal, spatial, and spatiotemporal activity patterns (*Kriegeskorte et al., 2008*). Instead of calculating correlations between the neural phenomena directly, this approach calculates the similarity in activity patterns between all possible trial combinations, resulting in a neural similarity matrix for each phenomenon with as many rows and columns as there are trials. Given that the bottom-left and top-right entries are identical in these matrices, we extracted only the top right entries excluding the diagonals containing auto-correlations and converted these entries into vectors. Next, second-order (Spearman) correlations of these trial-by-trial representational similarity vectors quantified the similarity in sensitivity to sensorimotor demands between all combinations of neural phenomena. This approach abstracts away from the activity patterns themselves such that similarities in sensitivity to sensorimotor demands across different movements between temporally or spatially non-overlapping neural phenomena can still be revealed. As above, the reliability of these representational similarities across the sample population was assessed using one-sample t-tests.

## Traveling wave analysis

Alpha and beta traveling waves were identified as cortical signals showing systematic phase variation across the electrode array (*Ermentrout and Kleinfeld, 2001*; *Muller et al., 2018*). We filtered the time-domain data with a two-pass third-order zero-phase shift Butterworth at individual alpha and beta frequency ranges determined using the four-step procedure outlined above. We applied the Hilbert transform to extract the instantaneous phase of ongoing rhythmic activity at each electrode and estimated for each instance of time (every ~2 ms) the spatial phase gradient across the

recording array. These spatial gradients represent distance-weighted phase shifts between cortical signals at neighboring recording electrodes, where positive phase shifts correspond to signals that have covered a greater distance along the unit circle and thus lead the oscillation (*Berens, 2009*). To quantify traveling wave direction and velocities along the cortical sheet, we projected and interpolated the phase data onto a two-dimensional plane defined by the first two principal axes of the electrode array. This approach facilitates visualization and interpretation of the subsequent gradient data and allows aggregating non-equidistant electrodes from adjacent grid and strip arrays. Wave directionality was then found by calculating the angle between spatial gradients estimated in both principal directions (1 cm in each direction). Wave velocity was found by the ratio between the mean frequency of the rhythm and gradient magnitude. To visualize the mean spatial progression of rhythmic activity across the electrode array, we subtracted the instantaneous phase at a central sensorimotor reference electrode from each electrode before averaging across trials and trial-time. We visualized the sample mean traveling wave direction by projecting and averaging over each participant's probability distribution of traveling wave directions onto the brain sagittal plane.

To assess whether the sensorimotor spatial gradients behaved like propagating waves at the single-trial level, we computed the phase-gradient directionality (PGD) across all stimulation-positive electrodes. PGD measures the degree of phase gradient alignment across an electrode array, taking a range of values between 0 and 1, and is found by the ratio between the norm of the mean spatial gradient and the mean gradient norm across the array (*Rubino et al., 2006*). We assessed the reliability of the propagating waves by finding the mean PGD across trials and trial-time and then comparing this value with two separate distributions of PGDs estimated from randomly permuted timepoints and randomly permuted electrode locations within the array. The former redistributes activity over time, preserving the spatial structure of activity in sensorimotor cortex, whereas the latter redistributes activity over space, preserving the temporal structure of activity in a trial. Rayleigh tests of uniformity were used to determine whether the traveling sensorimotor waves moved in a consistent direction across trials and trial-time (*Fisher, 1995*). To assess the consistency of wave propagation direction at a given time and electrode, we computed the directional consistency (DC). DC measures the degree of consistency in phase gradient direction, taking a range of values between 0 and 1, and is found by the mean resultant vector length across trials (*Zhang et al., 2018*).

## Data and code availability

Analysis code for spectral features extraction from the electrophysiological data are published as *Source code 1*.

## Acknowledgements

The authors thank the patients and their families for their participation and A Bastos, CW Hoy, and R Oostenveld for invaluable discussions and comments on previous versions of this article. AS was supported by Rubicon grant 446-14-007 from NWO and Marie Sklodowska-Curie Global Fellowship 658868 from the European Union. LB was supported by Brain and Cognition grant 433-09-248 from NWO awarded to IT.

## Additional information

### Competing interests

Floris P de Lange: Senior editor, *eLife*. The other authors declare that no competing interests exist.

### Funding

| Funder | Grant reference number | Author |
|---|---|---|
| Nederlandse Organisatie voor Wetenschappelijk Onderzoek | 446-14-007 | Arjen Stolk |
| European Commission | 658868 | Arjen Stolk |
| Nederlandse Organisatie voor Wetenschappelijk Onderzoek | 433-09-248 | Ivan Toni |

The funders had no role in study design, data collection and interpretation, or the decision to submit the work for publication.

## Author contributions

Arjen Stolk, Conceptualization, Software, Formal analysis, Investigation, Visualization, Methodology, Writing—original draft, Writing—review and editing; Loek Brinkman, Conceptualization, Data curation, Formal analysis, Investigation, Writing—original draft, Project administration, Writing—review and editing; Mariska J Vansteensel, Erik Aarnoutse, Frans SS Leijten, Chris H Dijkerman, Robert T Knight, Writing—review and editing; Floris P de Lange, Conceptualization, Writing—review and editing; Ivan Toni, Conceptualization, Formal analysis, Funding acquisition, Writing—original draft, Writing—review and editing

## Author ORCIDs

Arjen Stolk https://orcid.org/0000-0003-3798-4923
Floris P de Lange https://orcid.org/0000-0002-6730-1452

## Ethics

Human subjects: All participants had normal hearing and normal vision, and gave informed consent according to institutional guidelines of the local ethics committee (Medical Ethical Committee of the University Medical Center Utrecht, reference number 12-075), in accordance with the declaration of Helsinki.

## Decision letter and Author response

Decision letter https://doi.org/10.7554/eLife.48065.020
Author response https://doi.org/10.7554/eLife.48065.021

## Additional files

### Supplementary files

• Source code 1. Analysis code for the extraction of spectral features from the electrophysiological signal.
DOI: https://doi.org/10.7554/eLife.48065.015

• Transparent reporting form
DOI: https://doi.org/10.7554/eLife.48065.016

### Data availability

To preserve participant anonymity, raw patient data is available on request. Key analysis code has been uploaded as supplemental data and shared through the open-source FieldTrip toolbox (www.fieldtriptoolbox.org).

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

# Appendix 1

DOI: https://doi.org/10.7554/eLife.48065.017

## Supplemental analyses

Several control analyses were performed to test for alternative interpretations of the findings obtained with the IRASA technique and the $1/f$ slope index. First, the main analysis considering spectral features obtained using the IRASA technique revealed uncorrelated alpha and beta rhythmic activity in sensorimotor cortex. We performed an additional analysis testing whether power in the two frequency bands is also uncorrelated when broadband $1/f$ components of the signal are not accounted for, that is using the original power-spectra. It can be seen from the leftmost bars in *Figure 3—figure supplement 1* that performing the same correlation analysis on the original power-spectra yielded strong temporal and spatial correlations between alpha- and beta-band power. This observation underscores the importance of accounting for shared variance in alpha and beta power envelopes originating from broadband $1/f$ modulations. Second, the main analysis investigating the influence of rhythmic activity on local excitability found that the slope of the arrhythmic $1/f$ component had a differential relationship with alpha and beta rhythmic activity during movement imagery. It could be argued that the relation between beta rhythmic activity and the $1/f$ slope was artificially stronger because of the beta-band being closer than the alpha-band to the 30–50 Hz band of the power-spectrum on which the $1/f$ slope index is based. Accordingly, we performed an additional analysis grounded on the idea that a spurious interaction between beta-band power and the steepness of the $1/f$ slope should be amplified when both spectral features are directly based on the same (rhythmic) component of the power-spectrum, resulting in stronger correlations. As can be seen from *Figure 3—figure supplement 2*, correlations between beta-band power and the steepness of the $1/f$ slope were substantially reduced with both features based on the same component, compared to the original correlations shown in *Figure 3*. This observation indicates that the reciprocal changes between beta rhythmic activity and the slope of the arrhythmic $1/f$ component cannot be readily explained by a spurious relationship between these two spectral features.

Several other control analyses were performed to examine further the robustness and functional relevance of alpha and beta traveling waves. First, it could be argued that the traveling wave analyses depended on relatively noise-sensitive instantaneous estimates of phase and subsequent circular statistics. Accordingly, we performed an additional analysis that considered the entire time-series of alpha and beta rhythmic activity during movement imagery. Following insight from our phase-based analyses, showing activity moving along a rostro-caudal direction across the frontoparietal cortex, we calculated amplitude-based cross-correlations between electrode pairs aligned with the rostro-caudal axis in two representative individuals (see the brain insets in *Figure 4—figure supplement 1*). We rejected electrode pairs with cross-correlation functions explaining less than 50% of the mean distribution of cross-correlation in the sensorimotor cortex, based on leave-one-out cross-validation (1 and 3 alpha-band cross-correlation functions were held out in participants S1 and S2, respectively). This analysis showed that rostral electrodes led caudal electrodes in the alpha frequency range (red lines in *Figure 4—figure supplement 1*), consistent with alpha waves traveling in a rostral direction. Conversely, caudal electrodes led rostral electrodes in the beta frequency range (blue lines in the same figure), consistent with beta waves traveling in a caudal direction ($p<0.001$ for all lags, estimated from shuffled data using one-sample $t$-tests). This pattern of directionality is consistent with the instantaneous phase-based representations in *Figure 4*, showing concurrent alpha and beta waves traveling along opposite directions during movement imagery. Second, we examined whether the task-relevant traveling waves were additionally sensitive to selection demands during movement imagery. To this end, we examined the directional consistency (DC) of those waves, which measures the degree of consistency across trials in the phase-gradient direction. In the main analysis, it was found that alpha waves traveled more consistently through alpha-band local maxima ipsilateral to the

selected arm during movement imagery, as compared to baseline levels (*Figure 4D*). As seen in *Figure 4—figure supplement 2*, alpha waves propagated even more consistently through alpha-band local maxima during imagined movement of high demand trials, as compared to low demand trials. This effect occurred around the same time as alpha-band power increased during imagined movement of the ipsilateral arm (*Figure 2E*), particularly when selection demands were high (*Figure 2—figure supplement 3*). Taken together, these additional observations are consistent with the main findings of the study. Alpha and beta rhythm-dependent (dis)inhibition is task-relevant and propagated in a consistent spatiotemporal pattern.

