## [Decision Letter]

Thank you for submitting your article "Electrocorticographic dissociation of alpha and beta rhythmic activity in the human sensorimotor system" for consideration by *eLife*. Your article has been reviewed by three peer reviewers, one of whom is a member of our Board of Reviewing Editors, and the evaluation has been overseen by a Reviewing Editor and Laura Colgin as the Senior Editor. The following individuals involved in review of your submission have agreed to reveal their identity: Michael Breakspear (Reviewer #2).

The reviewers have discussed the reviews with one another and the Reviewing Editor has drafted this decision to help you prepare a revised submission.

Summary:

All reviewers and the editors agreed about the timeliness and relevance of your study. However, there were several concerns raised, which are summarized below. We are aware that these may be challenging to address within the usual timeframe; but we all agreed that if these issues can be addressed satisfactorily, the paper would substantially improve.

Essential revisions:

1) The overall link with behaviour seems weak, or absent altogether from some analyses. For example, there appears to be little connection between the wave analyses and the task; so, while interesting in itself, their relevance for the present paper is less clear.

Participants imagined either pronation or supernation in order to visualize the grasp – but this dichotomy was not used in the analysis, nor was the psychophysical estimate of what participants did, beyond a left/right comparison.

A revision should address what the functional relevance of the observed signals might be. For example, there are the ambiguous trials for which left/right answers are (roughly) equi-probable. Does this mean, for example, that both hemispheres now display similar activity changes during imagery, or that for the 50% crossing points, the brain responses are similar despite being at different points of stimulus orientation? Second, there are clearly no-brainer vs difficult trials. Does alpha/beta scale with this difficulty? Third, while not instructed, does the speed of the response relate to easy vs difficult trials, and/or to alpha/beta? Fourth, re-analysis epoched according to the motor response period rather than to the imaginary motor period of the task might help to establish commonalities/differences between the alpha/beta signals, which currently appear in rather similar profiles (Figure 2E).

You may conceive other analyses may be suitable to address this issue. Overall, however, without such a link, the paper seems to somewhat overstate its relevance, and makes interpretation of the specific neural changes difficult.

2) There was general question of how the results go beyond a descriptive outlining of power differences across areas of the cortex. For example, statements that the presence of a non-significant difference is evidence of no difference between alpha and beta topography would benefit from formal statistical tests of such differences.

It is difficult to see distinctly different temporal dynamics of power changes in the alpha and beta rhythms aggregated across the relevant local maxima as depicted in Figure 2E: Both show a peak increase in the ipsilateral side just after 1 second and a relatively smaller increase in the contralateral side at about the same time. The contralateral side changes are also comparable although the specific power changes differ. For the claim to be strongly made as it is in on related pages, a more formal statistical test of difference ought to be performed. But even when these analyses are performed in a more fine-grained manner (Figure 3A), statements that the presence of a non-significant difference is evidence of no difference seem to succumb to a logical fallacy. For this to be inferred, one ought to determine an effect large enough to be "important" and use this to perform an appropriate equivalence test. The same comment is true of the analyses using representational similarity (Figure 3D-F).

Additionally, if the signals are measured from different electrodes/regions, and are expressed in different frequencies, it may not be surprising that probing these regions has different effects; it wasn't entirely clear to us how the different comparisons were formally conducted. While the reviewers somewhat differed in their appraisal of the alpha/beta separation, they agreed upon discussion that the partial separation of alpha/beta is not completely inconsequential, but probably overstated, more so given the lack of relationship to functional behaviour. Similar concerns were raised about the representational similarity analysis and inferences about non-significant differences. For example, to which degree is the lack of similarity across alpha and beta confounded by the different anatomical sites from which they are recorded?

3) With regards to the concern about inferring E/I balance changes, 'regressing out' the background 1/f-like spectra has been conducted previously, but the reviewers shared skepticism about the merit in equating this unambiguously and directly to E/I balance. This is in part a semantic issue that can easily be corrected.

Additional control analyses, e.g. fitting a product of Lorentzians, may however provide additional assurance about the reliability of the result and the specific ranges and parameters chosen. The 30-50Hz range used here seems in line with some previous work (Gao et al., 2017), but where prior work was done in animals and with different recording techniques, it would be reassuring to see these parameters remain robust. Would it be possible to show such additional support (e.g. the individual fits/slopes and how reliable these are)?

4) All agreed that the wave analyses are novel and would provide important insight into the neural mechanisms underpinning imagery. One concern about the "wave analysis" is that showing a bias in a Hilbert-derived local phase vector is a bit of a zero-th order analysis. Fitting streamlines of wave propagation to spatially sampled data would permit a more principled analysis of wave patterns and their sources + sinks [R1, T1].

Second, given that the creation of travelling waves can be created spuriously and circular statistics are highly sensitive – it would be sensible to check the travelling wave analysis with a control analysis using e.g. phase shuffled data, that preserves the power spectrum but redistributes activity over time. The simple linear directionality of the travelling waves seems somewhat at odds with other reports recently that show much more complex patterns of activity. As per point 1, it is not clear how these analyses relate to the observed behaviour.

Not all of these analysis suggestions may be deemed within the scope of this paper, in which case you may want to at least consider this approach (and the broader computational modelling of waves in cortex) in the subsection, "Interpretational issues".

---

## [Author Response]

Essential revisions:1) The overall link with behaviour seems weak, or absent altogether from some analyses. For example, there appears to be little connection between the wave analyses and the task; so, while interesting in itself, their relevance for the present paper is less clear.Participants imagined either pronation or supernation in order to visualize the grasp – but this dichotomy was not used in the analysis, nor was the psychophysical estimate of what participants did, beyond a left/right comparison.A revision should address what the functional relevance of the observed signals might be. For example, there are the ambiguous trials for which left/right answers are (roughly) equi-probable. Does this mean, for example, that both hemispheres now display similar activity changes during imagery, or that for the 50% crossing points, the brain responses are similar despite being at different points of stimulus orientation? Second, there are clearly no-brainer vs difficult trials. Does alpha/beta scale with this difficulty? Third, while not instructed, does the speed of the response relate to easy vs difficult trials, and/or to alpha/beta? Fourth, re-analysis epoched according to the motor response period rather than to the imaginary motor period of the task might help to establish commonalities/differences between the alpha/beta signals, which currently appear in rather similar profiles (Figure 2E).You may conceive other analyses may be suitable to address this issue. Overall, however, without such a link, the paper seems to somewhat overstate its relevance, and makes interpretation of the specific neural changes difficult.

The motor imagery task exploited the fact that certain orientations afforded both over- and underhand grasping, whereas other orientations afforded grasping in a single manner only due to biomechanical constraints of the hand (Figure 1C). In previous related work using MEG in healthy participants (Brinkman et al., 2014), we used these differences in movement selection demands to distinguish between the contributions of alpha- and beta-band rhythms. In that study, we found that as selection demands increased, alpha-band power increased in the sensorimotor cortex ipsilateral to the hand used for motor imagery, whereas beta-band power concurrently decreased in the contralateral sensorimotor cortex. We examined the current ECoG dataset for similar effects, although the patients recorded in this ECoG study performed a substantially lower number of trials than the participants of the MEG study (120 vs. 480 trials, respectively). We defined high demand trials as trials involving cylinders oriented around the switch points estimated from each hand’s response curve, i.e. the 50% crossings in the response curves shown in Figure 1C (range: three orientation bins per switch point, i.e. -24° to +24°). We compared alpha- and beta-band temporal dynamics on high demand trials with those on low demand trials, defined as trials with cylinder orientations orthogonal to the switch points and covering an equivalent range. It can be seen from the new Figure 2—figure supplement 3, that the direction of the effects is consistent with the previous MEG observations (cf. Figure 2E-F of Brinkman et al., 2014). In the current ECoG study, there was a statistically significant decrease in contralateral beta rhythmic activity with increasing demand. However, the numerical increase in ipsilateral alpha rhythmic activity did not pass the statistical threshold. We now report this observation in the Results section to illustrate the consistency of this study with our previous work showing a functional dissociation based on selection demand, and to emphasize how the electrophysiological findings of this study are functionally related to participants’ behavior. More precisely, this modulation by task demand is consistent with the local cortical effects of alpha and beta rhythms found in this study, suggesting a role for alpha rhythms in inhibiting task irrelevant cortical regions and for beta rhythms in gating movement computations of neuronal populations in motor cortex. The limited number of trials available for analysis of task demands, however, raise concerns on the reliability of these findings. Accordingly, we refrained from using the effects of task demands for further analyses (apart from the additional wave analyses below).

“The temporal dynamics of these power changes are highly consistent with previous observations obtained from non-invasive electrophysiological recordings over sensorimotor cortex during performance of the same task, cf. Figure 3 in (Brinkman et al., 2014). In that magnetoencephalography (MEG) study, it was observed that as selection demands increased (when cylinder orientations afforded both over- and underhand grasping), alpha-band power increased in the sensorimotor cortex ipsilateral to the hand used for motor imagery, whereas beta-band power concurrently decreased in the contralateral sensorimotor cortex. We examined the alpha- and beta-band local maxima for similar effects, although the patients recorded in this ECoG study performed a substantially lower number of trials than the healthy participants of the MEG study (120 vs. 480, respectively). We defined high demand trials as trials involving cylinders oriented around the switch points in each hand’s response curve (range: three orientation bins per switch point, i.e. -24° to +24°). We compared alpha- and beta-band temporal dynamics on high demand trials with those on low demand trials, defined as trials with cylinder orientations orthogonal to the switch points and covering an equivalent range. It can be seen from Figure 2—figure supplement 3 that the direction of the effects is consistent with the previous MEG observations. There was a statistically significant decrease in contralateral beta rhythmic activity with increasing demand. However, the increase in ipsilateral alpha rhythmic activity did not pass the statistical threshold and concerns regarding the limited number of trials refrained us from using the effects of task demand for further analyses.”

It should be mentioned that application of the above analyses to the response period would not be informative given that, during this period, movement selection demands, effector type, as well as somatosensory reafference signals were not experimentally manipulated or controlled for. For instance, the response-related effects involve simple finger button presses and are not functionally related to the imagined arm-hand movements. For the same reason, response-locked analyses are also less informative than the electrical stimulation maps used for localizing functional territories along the central sulcus, despite being in general agreement with those territories (see Author response image 1, showing button press-related changes in alpha, beta, and high-frequency band power). We believe these response maps do not help to increase the focus and clarity of the manuscript, but are willing to include them if the editors/reviewers decide otherwise.

The reviewers also raised the issue of limited connection between the traveling waves and task performance. We analyzed the traveling waves for sensitivity to task demands. Following previous reports (e.g, Zhang et al., 2018), we examined the directional consistency (DC) of those waves, which measures the degree of consistency across trials in the phase-gradient direction. In our previous analyses, it was found that alpha waves traveled more consistently through alpha-band local maxima ipsilateral to the selected arm during imagined movement, as compared to baseline levels. As seen in Figure 4—figure supplement 2, the present analyses found that alpha waves propagated even more consistently through alpha-band local maxima during imagined movement on high demand trials, as compared to low demand trials. This effect occurred around the same time as alpha-band power increased during imagined movement of the ipsilateral hand (Figure 2E), particularly when selection demands were high (Figure 2—figure supplement 3). Taken together, these additional observations are consistent with the main findings of the study. Alpha and beta rhythm-dependent (dis)inhibition is task-relevant, and propagated in a consistent spatiotemporal pattern. We report this control analysis in the Supplemental Analyses.

“… Second, we examined whether the task-relevant traveling waves were additionally sensitive to selection demands during movement imagery. To this end, we examined the directional consistency (DC) of those waves, which measures the degree of consistency across trials in the phase-gradient direction. In the main analysis, it was found that alpha waves traveled more consistently through alpha-band local maxima ipsilateral to the selected arm during movement imagery, as compared to baseline levels (Figure 4D). As seen in Figure 4—figure supplement 2, alpha waves propagated even more consistently through alpha-band local maxima during imagined movement of high demand trials, as compared to low demand trials. This effect occurred around the same time as alpha-band power increased during imagined movement of the ipsilateral hand (Figure 2E), particularly when selection demands were high (Figure 2—figure supplement 3). Taken together, these additional observations are consistent with the main findings of the study. Αlpha and beta rhythm-dependent (dis)inhibition is task-relevant, and propagated in a consistent spatiotemporal pattern.”

2) There was general question of how the results go beyond a descriptive outlining of power differences across areas of the cortex. For example, statements that the presence of a non-significant difference is evidence of no difference between alpha and beta topography would benefit from formal statistical tests of such differences.It is difficult to see distinctly different temporal dynamics of power changes in the alpha and beta rhythms aggregated across the relevant local maxima as depicted in Figure 2E: Both show a peak increase in the ipsilateral side just after 1 second and a relatively smaller increase in the contralateral side at about the same time. The contralateral side changes are also comparable although the specific power changes differ. For the claim to be strongly made as it is in on related pages, a more formal statistical test of difference ought to be performed. But even when these analyses are performed in a more fine-grained manner (Figure 3A), statements that the presence of a non-significant difference is evidence of no difference seem to succumb to a logical fallacy. For this to be inferred, one ought to determine an effect large enough to be "important" and use this to perform an appropriate equivalence test. The same comment is true of the analyses using representational similarity (Figure 3D-F).Additionally, if the signals are measured from different electrodes/regions, and are expressed in different frequencies, it may not be surprising that probing these regions has different effects; it wasn't entirely clear to us how the different comparisons were formally conducted. While the reviewers somewhat differed in their appraisal of the alpha/beta separation, they agreed upon discussion that the partial separation of alpha/beta is not completely inconsequential, but probably overstated, more so given the lack of relationship to functional behaviour. Similar concerns were raised about the representational similarity analysis and inferences about non-significant differences. For example, to which degree is the lack of similarity across alpha and beta confounded by the different anatomical sites from which they are recorded?

We concur that formal tests are needed to support our statement regards the difference in temporal dynamics presented in Figure 2E. We thank the reviewers for raising this point and now report these statistics as well as formal evidence for the reviewers’ accurate observation that besides their differences, those temporal dynamics are on average highly correlated.

“We considered the temporal dynamics of power changes in alpha- and beta-band rhythms, aggregated across the relevant local maxima. These temporal dynamics were highly correlated (r = 0.7 ± 0.1, M ± SEM, p < 0.002) and both alpha- and beta-band power was more strongly attenuated for the hemisphere contralateral to the hand used in the imagined movement, see Figure 2E. Yet it can be seen from the same graph that alpha-band power increases in the (postcentral) cortex ipsilateral to the hand used for imagery, as compared to baseline levels (+34% between 910 and 1220 ms, p < 0.05; alpha-band power also decreased by 26% and 32% in the contralateral cortex between 170 and 850 ms and between 1230 and 2000 ms, respectively). In contrast, beta-band power decreases further in the (pre- and postcentral) contralateral cortex (-21% between 150 and 760 ms vs. -13% in the ipsilateral cortex between -180 and 580 ms; there was another statistically significant change of -21% from baseline in the contralateral cortex between 1450 and 2000 ms).”

This text illustrates how alpha- and beta-band rhythmic activity differ as a function of the effector used for motor imagery (statistically significant effects in opposite directions from baseline). Prompted by the first issue raised in the editor’s summary, we now additionally show how alpha- and beta-band mean temporal dynamics can be further dissociated on the basis of selection demand (our response to #1, Figure 2—figure supplement 2). Together, these observations demonstrate that even though alpha- and beta-band power changes are often correlated, their temporal dynamics can be functionally dissociated under the right circumstances.

We also agree with the reviewers that formal tests of no difference need to be performed. We have performed such tests using Bayesian statistics (using the JASP statistical software package) and now report Bayes Factors (BF_01_) for statistical tests evaluating evidence in favor of the null hypothesis, namely the relative likelihood of the data under the models at hand. All BF_01_s were larger than 1.56 in favor of the null hypothesis for the three tests of alpha-beta correlations shown in Figure 3A-C. This statistical evaluation favors the notion that the rhythms are uncorrelated (BF_01_ > 1) to the notion that they are correlated (BF_01_ < 1). The BF_01_s corresponding to the representational similarity analyses reported in Figure 3D-F were centered around 1. Each of these evaluations is roughly in agreement with alpha and beta rhythms having no substantial trial-by-trial correspondences (in sensitivity to sensorimotor demands), and the fact that all three tests do not provide strong evidence for either the null or alternative hypothesis adds additional weight to this conclusion.

A related concern raised by the reviewers is whether this lack of similarity in sensitivity to sensorimotor demand is confounded by the different anatomical sites from which alpha and beta rhythms are recorded. This is not the case given that both the correlation and representational similarity analyses considered alpha- and beta-band local maxima (Figure 3A,D) as well as alpha- and beta-band mean spatial and spatiotemporal patterns across the same functionally demarcated anatomical sites (Figure 3B,C,E,F). The latter two types of analyses thus considered alpha and beta rhythms at the same anatomical sites, whereas the former type considered alpha and beta rhythms at their local maxima. All these analyses “failed” to find evidence for a substantial relationship between alpha and beta rhythms considering their single-trial patterns as well as across-trial similarities, taking advantage of the high signal-to-noise ratio of ECoG (e.g., see Figure 2—figure supplement 2). Furthermore, elsewhere in the manuscript sensorimotor alpha and beta rhythms could be dissociated based on fine-grained anatomy, functional responses to electrical stimulation, movement selection dynamics, influence on local excitability, and propagation direction. We therefore believe that the separation of alpha and beta rhythmic activity in this manuscript is not partial nor overstated.

3) With regards to the concern about inferring E/I balance changes, 'regressing out' the background 1/f-like spectra has been conducted previously, but the reviewers shared skepticism about the merit in equating this unambiguously and directly to E/I balance. This is in part a semantic issue that can easily be corrected.Additional control analyses, e.g. fitting a product of Lorentzians, may however provide additional assurance about the reliability of the result and the specific ranges and parameters chosen. The 30-50Hz range used here seems in line with some previous work (Gao et al., 2017), but where prior work was done in animals and with different recording techniques, it would be reassuring to see these parameters remain robust. Would it be possible to show such additional support (e.g. the individual fits/slopes and how reliable these are)?

The relationship between the 1/f slope and local excitability is supported by the computational and empirical work of (Gao et al., 2017), and by recent work in humans indicating that the range from 30 to 50 Hz best predicts depth of sleep and of anesthesia, more so even than slow oscillatory power (Lendner et al., 2019 – now cited in the manuscript). We agree with the reviewers that caution is warranted due to the indirect link between 1/f slope and excitation/inhibition balance. Accordingly, we have changed our wording to reflect this caution, e.g. referring to “putative index” or “suggested link”.

To provide additional support for the reliability of the slope estimations, we now report their goodness of fit in the Methods section.

“Linear fits were used to estimate the steepness of the slope in the 30 – 50 Hz range (mean R^2^ across all slope fits in each individual = 0.95 ± 0.00).”

These numbers indicate that slopes of arrhythmic components could be accurately estimated with linear models (in log-log space), with 95% explained variance on average for all data segments on which power-spectra were estimated. This is further seen in the straight lines providing a good description of the individual power-spectral slopes shown in Figure 2—figure supplement 1.

We interpret the statement about fitting Lorentzians as referring to the IRASA technique we used here. Unlike fitting operations that depend on the specification and tweaking of parameters, this technique is purely data-driven. In IRASA, the only relevant parameters are the resampling factors, which are not meant to be tweaked. Accordingly, we used the default resampling factors as per the original report. For the source code, please see the ft_specest_irasa.m function in FieldTrip: https://github.com/fieldtrip/fieldtrip/blob/master/specest/ft_specest_irasa.m. As seen in Figure 2—figure supplement 1, this produced frequency ranges that showed a strong correspondence to the visible peaks in the original power-spectra (i.e. gray graphs in the inserts).

4) All agreed that the wave analyses are novel and would provide important insight into the neural mechanisms underpinning imagery. One concern about the "wave analysis" is that showing a bias in a Hilbert-derived local phase vector is a bit of a zero-th order analysis. Fitting streamlines of wave propagation to spatially sampled data would permit a more principled analysis of wave patterns and their sources + sinks [R1, T1].Second, given that the creation of travelling waves can be created spuriously and circular statistics are highly sensitive – it would be sensible to check the travelling wave analysis with a control analysis using e.g. phase shuffled data, that preserves the power spectrum but redistributes activity over time. The simple linear directionality of the travelling waves seems somewhat at odds with other reports recently that show much more complex patterns of activity. As per point 1, it is not clear how these analyses relate to the observed behaviour.Not all of these analysis suggestions may be deemed within the scope of this paper, in which case you may want to at least consider this approach (and the broader computational modelling of waves in cortex) in the subsection, "Interpretational issues".

We are grateful for these suggestions which motivated us to perform additional traveling wave analyses that did not depend on instantaneous phase estimates (in a similar spirit to the suggested reports). Following insight from our phase-based analyses, showing activity moving along a rostro-caudal direction across the frontoparietal cortex, we calculated amplitude-based cross-correlations between electrode pairs aligned with the rostro-caudal axis in two representative individuals (see the brain insets in Figure 4—figure supplement 1). This analysis considered entire time-series of alpha and beta rhythmic activity during movement imagery and produced observations consistent with the phase-based analyses, while avoiding to make specific assumptions on wave patterns. Namely, rostral electrodes led caudal electrodes in the alpha frequency range (red lines in Figure 4—figure supplement 1), consistent with alpha waves traveling in a rostral direction. Conversely, caudal electrodes led rostral electrodes in the beta frequency range (blue lines), consistent with beta waves traveling in a caudal direction (p < 0.001 for all lags). This pattern is consistent with the instantaneous phase-based representations in Figure 4, showing concurrent alpha and beta waves traveling along opposite directions during movement imagery. We report this control analysis in the Supplemental Analyses.

“Several other control analyses were performed to further examine the robustness and functional relevance of alpha and beta traveling waves. First, it could be argued that the traveling wave analyses depended on relatively noise-sensitive instantaneous estimates of phase and subsequent circular statistics. Accordingly, we performed an additional analysis that considered the entire time-series of alpha and beta rhythmic activity during movement imagery. Following insight from our phase-based analyses, showing activity moving along a rostro-caudal direction across the frontoparietal cortex, we calculated amplitude-based cross-correlations between electrode pairs aligned with the rostro-caudal axis in two representative individuals (see the brain insets in Figure 4—figure supplement 1). We rejected electrode pairs with cross-correlation functions explaining less than 50% of the mean distribution of cross-correlation in the sensorimotor cortex, based on leave-one-out cross-validation (1 and 3 alpha-band cross-correlation functions were held out in participants S1 and S2, respectively). This analysis showed that rostral electrodes led caudal electrodes in the alpha frequency range (red lines in Figure 4—figure supplement 1), consistent with alpha waves traveling in rostral direction. Conversely, caudal electrodes led rostral electrodes in the beta frequency range (blue lines in the same figure), consistent with beta waves traveling in a caudal direction (*p* < 0.001 for all lags, estimated from shuffled data using one-sample *t*-tests). This pattern of directionality is consistent with the instantaneous phase-based representations in Figure 4, showing concurrent alpha and beta waves traveling along opposite directions during movement imagery. Second, …”